# The jingle fallacy in comprehension tests for reading

**Charlotte E. Lee** [ID] *, **Hayward J. Godwin, Denis Drieghe**

School of Psychology, University of Southampton, Southampton, United Kingdom

* c.lee@soton.ac.uk

**Data Availability Statement:** All data and materials are available online at: https://osf.io/dk82u/?view_only=88d54bdc1f9b4793ba4b2dfebf106085.

**Funding:** CL was funded by the UKRI Economic and Social Research Council South Coast Doctoral Training Partnership (Grant Number ES/P000673/1). This work formed part of a PhD with the South

## Abstract

The *Jingle fallacy* is the false assumption that instruments which share the same name measure the same underlying construct. In this experiment, we focus on the comprehension subtests of the Nelson Denny Reading Test (NDRT) and the Wechsler Individual Achievement Test (WIAT-II). 91 university students read passages for comprehension whilst their eye movements were recorded. Participants took part in two experimental blocks of which the order was counterbalanced, one with higher comprehension demands and one with lower comprehension demands. We assumed that tests measuring comprehension would be able to predict differences observed in eye movement patterns as a function of varying comprehension demands. Overall, readers were able to adapt their reading strategy to read more slowly, making more and longer fixations, coupled with shorter saccades when comprehension demands were higher. Within an experimental block, high scorers on the NDRT were able to consistently increase their pace of reading over time for both higher and lower comprehension demands, whereas low scorers approached a threshold where they could not continue to increase their reading speed or further reduce the number of fixations to read a text, even when comprehension demands were low. Individual differences based on the WIAT-II did not explain similar patterns. The NDRT comprehension test was therefore more predictive of differences in the reading patterns of skilled adult readers in response to comprehension demands than the WIAT-II (which also suffered from low reliability). Our results revealed that these different comprehension measures should not be used interchangeably, and researchers should be cautious when choosing reading comprehension tests for research.

## Introduction

Reading comprehension is a complex task made up of interactions between the features of a text and the skill and strategies of the reader [1–3]. The Simple View of Reading [4] describes the basic requirements for reading as the ability to decode and identify words in text by converting graphemes into phonemes combined with the ability to understand information presented orally (language comprehension). However, in the more complex Construction-Integration (CI) model of reading comprehension [1], text is represented by a surface structure

Coast Doctoral Training Partnership (https://southcoastdtp.ac.uk/). The funders had no role in study design, data collection and analysis, decision to publish, or preparation of the manuscript.

**Competing interests:** The authors have declared that no competing interests exist.

(semantic representations of words within a text), a textbase (a representation of the explicit meaning of the whole text, coherently integrating each word meaning) [5], and a situation model (where a reader creates a model of the situation, integrating the explicit meaning of the text with their own world knowledge). For shallow comprehension, a textbase is sufficient, however for deeper understanding a situation model is required. Differences in theoretical conceptualisation of comprehension can result in differences in the underlying mechanisms measured by comprehension tests based upon them. Indeed, inconsistencies in research where skills measured by cognitive tasks are used to predict readers' performance on reading comprehension measures have been suggested to reflect differences in underlying cognitive mechanisms [6–9]. The current paper strives to shed some light on the problems that researchers may face when selecting reading comprehension tests, and the direct impact that test selection can have on conclusions based upon them in eye tracking investigations.

## Evidence for a jingle fallacy

In some of our previous eye movement investigations of average-to-very-skilled readers [10, 11] we found that two often-used reading comprehension subtests from standardised reading ability measures failed to load together in a principal components analysis and were only weakly correlated (r = 0.21, [10]; r = 0.15, [11]). These subtests were from the Wechsler Individual Achievement Test (WIAT-II UK [12]) and the Nelson Denny Reading Test (NDRT [13]). We concluded that these comprehension tests might be assessing different underlying skills. Since these measures are both named 'reading comprehension', this would present a clear example of Thorndike's *Jingle fallacy*: that is, the misleading assumption that two measures assess a single underlying construct because they share the same name [14]. Although not uncommon in psychological research, where a variety of tests are available to assess common constructs, problems when selecting and reporting appropriate measures can lead to questionable research practices when used for scientific purposes [15]. The aim of the current paper is to extend our previous investigations to directly test the differences between the two tests by using them to predict differences in eye movement patterns reflective of different comprehension demands and to further highlight the pitfalls of comparing research that uses either test for this area of research.

## Differences in test format

We start by discussing some qualitative differences in the format of the two comprehension tests that may provide some insight into the underlying constructs that are being tapped into by each one. First, the NDRT exclusively features non-fiction passages whereas the WIAT-II features more varied text formats, with some fiction and non-fiction passages as well as single sentences. Previous studies have noted that differences in the format of reading materials (sentences vs paragraphs [16], fiction vs non-fiction [e.g., 17–19]) can impact reading behaviour as reflected in eye movement measures. Reading times are longer and rereading is more common for sentences presented within paragraphs than for sentences presented alone, which suggests that text format influences the reading strategy used to comprehend the text [16]. Best et al. [17] also showed that comprehension accuracy was higher for narrative texts than expository texts (non-fiction/scientific) and performance on each was predicted by different individual skills. Decoding skills were a key element for successful narrative text comprehension, whereas world knowledge was more important for successful expository text comprehension. While this may suggest that narrative texts included in the WIAT-II where comprehension is suggested to be higher, might be 'easier' for skilled readers, it also suggests that a reliance on non-fiction passages in the NDRT may result in greater overlap with general knowledge. This was

also suggested by Ready et al. [20] following work by Coleman et al, [21] who found that college students could answer the questions on NDRT comprehension tests and achieve a greater-than-chance level of accuracy without actually reading the associated passages. However, we note that accuracy was 44–47% whereas chance level was 20% so the test is clearly measuring more than just general knowledge.

Both tests also feature explicit differences in reading instruction since the WIAT-II includes a combination of silent and oral reading, whereas the NDRT only features silent reading. Reading aloud involves articulating the text as well as the standard process of reading, and evidence from the eye-voice span (the distance between the location of a fixation and the articulated word) demonstrates that oral reading involves additional working memory processes [22]. Hale et al. [23] investigated differences in reading aloud and silently and found that for children across grades 4–12 reading comprehension was higher when reading aloud than when reading silently. In addition, some prior research suggests that changing oral and silent reading tasks in comprehension tests may lead to different outcomes, though this has been noted specifically in relation to differences between children with reading difficulties and average readers [24]. Much less is known about how comprehension changes when adults read aloud. A survey by Duncan and Freeman [25] reported that, of 529 respondents, 67.5% said that they read aloud to understand difficult text (though they noted that this was usually only brief). Gambrell and Heathington [26] reported that 36% of poor adult readers said that they could read more quickly when reading aloud compared to just 4% of good readers. It may be that an oral reading task to assess comprehension is less informative for adult readers due to individual differences, though more research is needed to investigate this.

Another notable difference is that testing in the WIAT-II is administered by an experimenter who asks questions aloud to the participant and records their spoken responses on paper, whereas the NDRT is administered independently. This procedural difference could lead to performance anxiety for participants when taking the WIAT-II and may introduce noise into data collected under these conditions. This may be especially important where participants are sometimes asked to read aloud. In contrast it may mean that the NDRT has comparatively less control to determine whether a participant is properly engaging with the task. This aspect highlights another qualitative difference in the administration of the WIAT-II in comparison to the NDRT.

A good comprehension test should be able to predict differences in behaviour between tasks that vary in comprehension demands. Eye movement measures reflect complex cognitive processes active during reading [27, 28]. The current paper therefore investigated global reading strategies for paragraph reading and aims to examine whether the differences we described between the comprehension subtests of the WIAT-II and the NDRT impact their ability to predict eye movement patterns reflecting changes in comprehension demands.

## Individual differences in adult readers' eye movements

We turn our focus now to individual differences in adult readers' eye movements. Skilled adult readers typically read more quickly, make fewer and shorter fixations, longer saccades and fewer regressions than less skilled readers [28, 29]. However, reading skill is not directly related to reading rate, and a speed-accuracy trade-off means that faster reading eventually leads to lower levels of comprehension [30]. There is much variability within groups of skilled adult readers with fixation durations varying between approximately 50–600 ms and saccade lengths between 1–20 letter spaces [31, 32]. Skilled readers also vary in how they respond to features of a text. Ashby et al. [29] found that poor adult readers (identified using NDRT reading comprehension and vocabulary tests) benefitted more from highly constraining sentential contexts

compared to skilled readers. Similarly, Bisanz et al. [33] reported a complex relationship between reading ability and reading times in line with Stanovich's [34] interactive-compensatory model which stated that poor readers, who had below average bottom-up processing skills, would rely more heavily on contextual cues when they were available. Bisanz et al. [33] showed that poor readers actually read some sentences more quickly than skilled readers. It has been suggested that some readers might use a 'risky' reading strategy where they read more quickly and make fewer refixations than other readers [35].

## Intra-individual differences

In addition to the differences observed between readers, intra-individual differences (variability within the same reader) can also influence reading behaviours. It has been well established that task demands can influence the way that readers process a text: skilled readers are able to adjust their reading behaviours (and pace) to the demands of the task [36] and are able to read thoroughly or superficially when needed [37]. Aaronson and Ferres [38, 39] noted that skilled readers are more likely to use a 'recall strategy' focussed on structural aspects of a text when a reading task involves direct recall of words/sentences, but when the task involves true/false questions, their focus is driven by the meaning of the text using a 'comprehension strategy'. This research was influential as it gave clear evidence that skilled readers had some autonomy over how deeply they processed a text.

It has been noted that when texts are more difficult, a more 'careful' strategy might be used where, in comparison to a risky reading strategy [35], readers tend to make more refixations, have smaller average fixation durations and smaller saccade amplitudes [40]. Researchers have investigated whether these strategies can be observed for identical sentences when different comprehension demands are placed upon them. Radach et al. [16] investigated differences in eye movement behaviours related to the specific reading task as well as different text formats. Participants took part in one of two tasks: comprehension, where participants were asked detailed questions about the text; and a word verification task where participants had to indicate which word had appeared in the sentence from some given options. Radach et al. [16] also compared eye movement measures within these groups for identical sentences that were either embedded within a passage or were presented alone. Researchers concluded that top-down processes influenced by the task (comprehension vs word identification) and format of the text (sentences vs paragraphs) clearly impacted the eye movement record. Word-viewing times were significantly longer on comprehension tasks and more fixations were made on a word in this task than in the verification task, indicating more careful reading when reading for comprehension. Passages were read more quickly on the first-pass but featured more rereading than sentences.

Similarly, Wotschack and Kliegl [41] investigated the effect of easy 'verification' questions (after 27% of sentences) compared to 'hard' comprehension questions about sentence meaning (following 100% of sentences) and found that the more difficult questions were associated with more careful reading as indicated by more rereading and more regressions. However, they found that accuracy was high in both conditions and questioned the strength of their manipulation. In response, Weiss et al. [42] aimed for a stronger difficulty manipulation and investigated 'easy' lexical verification questions versus 'difficult' comprehension questions that required resolving some syntactic ambiguity. For example, a sentence containing a subjective relative clause such as 'The chef that distracted the waiter sifted the flour onto the counter', was followed by an easy question: 'Did a chef do something?' Or a difficult question: 'Did the waiter distract the chef?' They did see differences in accuracy between the difficult (83%) and easy (97%) conditions, and also found that participants made more regressions and spent

more time rereading texts in the difficult condition but that no disruptions were seen in first pass fixation times. Weiss et al. [42] concluded that inflated differences happened at the end of passages even when the ambiguity occurred earlier in the sentence. Accuracy was not predicted by the magnitude of the disruption, suggesting that the increased processing time was a 'checking mechanism' rather than additional information processing.

Christianson et al. [43] reached a conclusion similar to Weiss et al. [42] in a study that investigated rereading behaviours in garden-path sentences (where an ambiguity in the sentence meaning is revealed fairly late in the sentence e.g. The babysitter who was purchased a gift card thanked the parents) vs. local coherence structures (where ambiguities were resolved earlier, e.g. The parents thanked the babysitter who was purchased a gift card). They found that rereading behaviours were more consistent with confirmatory rereading (checking) than revisionary rereading (for understanding) because rereading was not consistently predicted by critical regions in the sentence structure, and rereading behaviours were not predictors of off-line comprehension accuracy.

Recent investigations have looked more closely at rereading behaviours and have started to examine individual differences in rereading. A study by Andrews and Veldre [44] investigated 'wrap-up' effects in tasks with different comprehension loads in relation to individual differences in reading proficiency (measured by vocabulary, reading comprehension reading rate (NDRT [13]), spelling dictation and spelling recognition [45]). Wrap-up effects [46] are where longer reading times are observed at clause and sentence boundaries, where readers integrate information before moving forward in a text [47, 48]. Wrap-up times have been associated with the goals of the reading task, for example in a study by Stine-Morrow et al. [49] where differences in wrap-up predicted recall but not comprehension success. Importantly, Andrews and Veldre assessed readers' individual differences in spelling, reading comprehension (NDRT), vocabulary and reading rate alongside manipulating how often comprehension questions occurred (after 25% of passages or 100% of passages). They found that comprehension load had little effect on wrap-up, however it did lead to shallower (more risky) reading strategies when comprehension demands were low, with longer passage reading times, more refixations and regressions, but no differences in average fixation times or forward saccade lengths. Andrews and Veldre [44] found that the better readers (as identified via a composite score of the individual differences measures that have been shown to provide a good assessment of lexical quality [50–54]) generally read passages more quickly, made fewer and shorter fixations, longer forward saccades and marginally fewer regressions than poorer readers. They did not find that reading proficiency composite scores interacted with the effect of comprehension load on eye movement measures, but they noted that accurate comprehension was associated with more consistent reading behaviour, where readers did not adjust their reading strategy much in response to comprehension load.

Reading strategies may of course be adapted over time during an experiment. For example, readers may read through early trials more slowly when they have higher comprehension demands, until they are familiar with the format of the questions in the experimental block, after which they may adjust their reading rate to speed up processing time. This rate of adaptation may be modulated by individual differences, whereby better comprehenders might be able to increase their reading rate to one that is optimal/preferred more quickly over trials than less skilled comprehenders. Therefore, besides examining the differences between predictions based on two comprehension tests, a second goal of current study was to determine whether individuals alter their reading strategies in response to comprehension demands gradually as trials progress. We were interested to see if individual differences in reading ability predicted differences in the rate of adaptation to different comprehension demands as well as whether discrepancies occurred between the two measures of reading comprehension that we

included. Following Radach et al. [16], identical reading materials were used between conditions in the current study to directly compare the influence of comprehension demands placed on the reader via differences in the difficulty of questions that followed.

## Predictions

We expected high scores on the comprehension tests to predict faster passage reading times as faster sentence reading times were associated with higher scores on the comprehension subtests from the WIAT-II [12] and the NDRT [13] in Lee, Godwin et al. [10] and Lee, Pagán et al. [11]. Note however that the format of our experimental materials in the current study (paragraphs) was different in comparison to our previous investigations (sentences). Longer reading times and more rereading have been observed for passages compared to sentences [16]. Similarly, since comprehension was included as part of the composite measure of reading proficiency in Andrews and Veldre [44],who found that higher reading proficiency predicted faster passage reading times, shorter average fixation durations, longer forward saccades and a greater number of regressions than low proficiency, we expected similar patterns to emerge for our comprehension scores.

We expected that higher offline comprehension scores would predict faster passage reading times, shorter average fixation durations, longer forward saccades and fewer regressions. Higher comprehension demands were expected to increase the number of fixations and the time that participants spent reading the passages. We anticipated that all readers would adapt their reading strategy to become more efficient (they would make fewer fixations, longer saccades, shorter fixations and read passages more quickly), but that there might be individual differences observed in the rate of adaptation or ceiling levels in saccade lengths that poorer readers could reach, since poorer readers have been shown to have shorter rightwards perceptual spans (in languages read left to right) than better readers [55]. Similarly, as poorer readers usually exhibit slower reading times and longer fixations than skilled readers [10, 11, 28, 29, 44] we anticipated floor effects for poor readers' minimum passage reading times, fixation durations and the number of fixations. Since the intended population was skilled adult readers, it was likely that accuracy would be high across tasks (as was observed in [44, 41]). Therefore, because comprehension accuracy is often higher for narratives than expository texts [17], expository passages were used in the current study to maximise the likelihood of variability in accuracy scores.

We note that the NDRT exclusively uses expository texts to measure comprehension, therefore it may be more similar in format to the passages used in this experiment. As noted by Ready et al. [20] and Coleman et al. [21], the NDRT may also feature a high degree of overlap with general knowledge or world knowledge, which has been found to be associated with expository text comprehension. Therefore, it would not be surprising if the NDRT predicts higher comprehension accuracy across conditions, than the WIAT-II, which features more varied reading formats. We also anticipated that the WIAT-II may be more noisy in its predictions due to some performance anxiety induced by the experimenter's presence.

## Method

### Participants

Participants were 91 students and staff from the University of Southampton over the age of 18 (11 Males, M = 20.27 years, range = 18–45 years). An additional 9 participants took part in the study, but their data were removed from the final dataset due to poor overall accuracy on the comprehension questions in the eye tracking task (below 60% where chance level was 50%). Participants were all native English speakers with normal or corrected to normal vision and no

known reading difficulties. Participants received course credits or £25 for completing the study. Recruitment took place from 29/10/2021 to 10/06/2022. This study was approved by the University of Southampton Ethics and Research Governance Board.

## Apparatus

Paragraphs and questions were presented on a 21-inch CRT monitor, with a refresh rate of 120 Hz and a resolution of 1024 x 768 at a viewing distance of 60 cm. Passages were presented in Courier New, size 14 font on a grey background; three characters equated to about 1˚ of visual angle. Although reading was binocular, eye movements were recorded from the right eye only using an EyeLink 1000 tracker [56]. Forehead and chin rests were used to minimize head movements. The spatial resolution of the eye tracker was 0.05˚, and the sampling rate was 1000 hz.

Participants used a 14-inch Dell Laptop Computer to complete the NDRT comprehension test administered using an online web browser running Qualtrics. For copyright issues, whenever we ran a participant using the online version, we voided a purchased paper version. Participants were required to select answers using a mouse. During WIAT-II comprehension test researchers used the testing flip pad and scoring sheets included in the test pack.

## Materials

Forty experimental paragraphs (M = 138.33 words, SD = 19. 28) were adapted from freely available online practice comprehension tests [57]. Two conditions were created for each paragraph, one with lower comprehension demands where participants were asked 'What is the passage about?' and were given two short options that consisted of a word or phrase (e.g., Synaesthesia/Claustrophobia). One option was directly related to the passage and the other was unrelated. In the higher comprehension demands condition participants were asked, 'What is the main idea of the passage?' and two longer and more detailed options were presented from which participants were asked to select an answer (e.g., People with synaesthesia experience a fusing of different senses/People with synaesthesia may hear a sound when they touch an object). In this condition, both answers were related to the passage, but one provided a better evaluation of the passage meaning. Questions were similarly phrased but differences were presented by the type of options available, and level of detail needed to select a correct answer. The original questions from the online practice materials were the 'higher demands' questions, a 'lower demands' alternative was then created for each of them. Paragraph naturalness and comprehension question difficulties were independently rated by participants who did not take part in subsequent testing. Passages were rated on a scale from 0 (very unnatural) to 100 (very natural) (M = 63.04, SD = 5.31) to ensure there were no outliers in the readability of the text and questions were rated on a rated as more difficult (M = 23.71, SD = 4.45) than low comprehension demand questions (M = 19.97, SD = 3.67), t (49) = - 8.57, p < .001. Two counterbalanced lists were then created so that each participant viewed 20 of each question type but did not view the same paragraph twice. The paragraphs occupied 10–13 lines on the screen (M = 859.95 characters including spaces, Max = 1159 characters).

## Design and procedure

Testing took place over two sessions with a minimum of two days in between them. During the first session participants were given an information sheet and were asked to sign a consent form and completed two eye tracking tasks (the first eye tracking task was for a separate experiment, where participants read 60 single sentences and lasted approximately 30 minutes), followed by the experimenter administered WIAT-II comprehension test and some other

cognitive tasks belonging to an unrelated experiment (Rapid Automatized Naming and the pseudoword decoding and word reading subtests of the WIAT-II. These tasks took approximately 15 minutes to complete). The same experimenter administered this task to all participants to control for as much experimental variation between participants as possible. A script was read from the test materials to ensure that instructions were identical for all participants. Participants read passages (short narratives and information texts) aloud or silently and were asked literal and inferential comprehension questions by the experimenter, participants gave spoken responses which the experimenter transcribed.

For the eye tracking task, participants were asked to sit comfortably at the computer, resting their chin on a chinrest and were then guided through the set up and 9-point calibration of the eye tracker by the researcher. Participants were then required to direct their gaze to a fixation cross presented in the upper left portion of the screen. Once participants fixated upon the cross sentences were presented and always began at the location marked by the fixation cross. Participants were asked to read the paragraphs and answer questions presented on the screen using the keyboard to respond. Participants either answered questions with longer, and more detailed options from which to select an answer (higher comprehension demands) or with shorter, simpler options (lower comprehension demands) depending on the condition. The same participants completed both conditions over two sessions. Eye tracking sessions took place on two separate days. At the start of the blocks, participants read five practice paragraphs with questions matching the type for the current condition. Practice questions were followed by 20 experimental paragraphs that were each followed by a comprehension question. Block order was counterbalanced so that participants who read paragraphs and answered questions with higher comprehension demands in session 1, then read paragraphs and answered questions with lower comprehension demands in session 2 and vice versa. Block order was randomly assigned and within each block trial order was randomised. Participants could take breaks when needed.

During the second session participants took part in the second part of the eye tracking task (featuring the condition that they had not yet completed). Participants were then asked to complete the NDRT comprehension test, and some other online tasks for a separate study (the vocabulary subtest of the NDRT, a test of vocabulary knowledge, spelling dictation and spelling recognition tasks, an Author Recognition test, and a backwards digit span task in a randomised order. These tasks took approximately 40 minutes to complete). During the NDRT participants silently read up to 7 passages and answered 5–8 MCQ questions about them with 4 available options. Questions appeared below the passages on the same screen. On the first passage participants were stopped after 1 minute and were asked to record the number corresponding to the line that they had been reading to measure their reading rate. Testing automatically stopped after 10 minutes and answers were recorded. Testing for the NDRT comprehension test followed a half-timed procedure, in which the standard time limit for completing this test was reduced by half. Participants were not aware of this reduced time limit. This procedure has been shown to generate a more normal distribution for university student readers (like those who took part in the current study) than the standard time limits in an investigation by Andrews et al. [50]. To measure test reliability, Cronbach's alpha was calculated for both the WIAT-II comprehension test ($\alpha = .62$) and the NDRT comprehension test ($\alpha = .75$). We note that even though these estimates are still considered acceptable, they are lower than those reported for normed data (NDRT = .89 to .98 [12] and WIAT = .98 [13]). Given that we focus on a university sample of readers, this might suggest these comprehension tests are less reliable for this population of readers. Furthermore, we note that by using a timed version of the NDRT comprehension test, reliability will be lower for instances where participants answered fewer questions in the given time.

## Results

Overall accuracy on comprehension questions was high but not at ceiling level (M = 79.42%, SD = 10.53). Reading comprehension scores on WIAT-II were calculated and normed following guidance from the experimenter manual (M = 110.47, SD = 9.37, range = 71–124). NDRT comprehension scores were calculated based on raw scores due to the half-timed aspect of the task. NDRT comprehension scores (M = 57.64, SD = 11.02, range = 20–74) were weakly correlated with the WIAT-II comprehension scores (r = 0.22, p = .039). Both WIAT-II Comprehension and NDRT Comprehension were weakly correlated with overall accuracy on the eye tracking task (WIAT-II r = .22, $p < .001$; NDRT r = 0.11, $p < .001$). Scores on both tests were standardised for further analyses.

### Data cleaning

Eye tracking trials identified by the experimenter as having issues with tracker loss or featuring excessive blinking were removed prior to the analysis. Fixations shorter than 80 ms that landed within one character of the previous or next fixation were merged. Then, of the remaining fixations, those shorter than 80 ms and longer than 800 ms were removed. Practice trials were also removed. Due to an error in the programming of the experiment, texts were presented with a justified alignment which meant that word level data would have confounds between word length and visual extent. For this reason, word level measures such as regressions and refixations were not included in these analyses.

The following global eye tracking measures were calculated for each trial; Number of Fixations (total number of fixations made on a trial); Average Fixation Duration (mean duration in ms of all fixations in a trial); Forward Saccade Length (the distance in degrees of visual angle between one fixation and the next); and Total Passage Reading Time (total time in ms spent reading the passage in a trial). Trials where total passage reading times fell outside of 2.5 standard deviations from the mean for each participant were removed as outliers (1.31% of data removed). Data were then removed for each eye movement measure per participant that fell outside of 2.5 standard deviations from the mean (Number of Fixations (0.59% data removed); Average Fixation Duration (0.88% data removed); Forward Saccade Length (1.16% data removed). Descriptive statistics per condition for these measures were calculated across participants and are displayed in Table 1.

### Linear mixed models

Eye movement measures were analysed using the lme4 package (version 1.1–31 [58]) in R (version 4.2.2 [59]). Data were checked for normality and were not transformed for modelling as

**Table 1. Descriptive statistics for eye movement measures.**

| | Condition | Min | Max | Mean | SD |
|---|---|---|---|---|---|
| Number of Fixations | Low | 42.00 | 285.00 | 138.38 | 33.43 |
| | High | 65.00 | 269.00 | 143.64 | 33.95 |
| Average Fixation Duration (ms) | Low | 138.53 | 285.94 | 206.14 | 24.06 |
| | High | 144.30 | 279.17 | 206.79 | 23.80 |
| Average Forward Saccade Length (visual degrees) | Low | 3.38 | 9.66 | 6.07 | 0.98 |
| | High | 3.41 | 9.25 | 5.99 | 0.99 |
| Total Passage Reading Time (ms) | Low | 8388.00 | 58115.00 | 28854.00 | 8484.70 |
| | High | 11232.00 | 63869.00 | 30138.17 | 8868.98 |

Descriptive statistics are based on participant means per condition.

**Table 2. LMM for number of fixations predicted by NDRT comprehension and interactions with trial number and condition.**

| | β | 95% CI | t | df | p |
|---|---|---|---|---|---|
| Intercept | 153.47 | [147.01, 159.92] | 46.60 | 140.91 | < .001 *** |
| Trial Number | -0.75 | [-0.85, -0.64] | -14.38 | 3300.23 | < .001 *** |
| Condition | 9.64 | [5.20, 14.07] | 4.26 | 335.04 | < .001 *** |
| NDRT Comprehension | -1.10 | [-6.65, 4.46] | -0.39 | 104.33 | .699 |
| Trial Number × Condition | -0.22 | [-0.42, -0.01] | -2.09 | 3302.67 | .037 * |
| Trial Number × NDRT Comprehension | -0.16 | [-0.26, -0.06] | -3.13 | 3299.89 | .002 ** |
| Condition × NDRT Comprehension | -4.23 | [-8.60, 0.14] | -1.90 | 346.14 | .059. |
| Trial Number × Condition × NDRT Comprehension | 0.25 | [0.05, 0.45] | 2.41 | 3300.46 | .016 * |

The baseline of the condition term is lower comprehension demands. Estimates represent the change when going from lower to higher comprehension demands.

their distribution closely resembled a normal distribution. Binomial Generalized Linear Mixed Models were used to model accuracy data. The following model building strategy was followed. Models featured all fixed effects of interest: the main effect of experimental condition (lower vs higher comprehension demands), either the NDRT or WIAT-II comprehension test scores and the trial number and all the interactions. To ensure the maximal model was achieved, we started with a full random structure (all random slopes were included for subjects and items) and performed stepwise trimming of this structure until the model converged [60]. Slopes were first trimmed from the random effects structure where perfect correlations were indicated and subsequently factors that explained the smallest amount of variance until the model converged.

**Number of fixations.** Models shown in Tables 2 and 3 indicated that overall, more fixations were made on paragraphs where comprehension demands of the questions were high compared to when they were low. The number of fixations decreased slightly over trials, however, a significant three-way interaction between trial number, condition and scores on the NDRT comprehension test revealed a more complex pattern based on individual differences (Table 2). Fig 1 shows that high scorers on the NDRT comprehension test reduced the number of fixations further into the experimental session (analyses were based on continuous comprehension scores but are presented in 3 panels for the mean +/- 1SD in figures to clearly demonstrate the 3-way interaction). They also made more fixations in the difficult condition and these two factors did not interact. A different pattern emerged for the low scorers on the NDRT comprehension test. Low scorers did make more fixations on a paragraph at the beginning of the experiment than on trials nearing the end of the experiment, but when

**Table 3. LMM for number of fixations predicted by WIAT-II comprehension and interactions with trial number and condition.**

| | β | 95% CI | t | df | p |
|---|---|---|---|---|---|
| Intercept | 153.17 | [146.69, 159.64] | 46.35 | 140.84 | < .001 *** |
| Trial Number | -0.77 | [-0.87, -0.67] | -14.77 | 3302.63 | < .001 *** |
| Condition | 9.22 | [4.78, 13.65] | 4.07 | 351.96 | < .001 *** |
| WIAT-II Comprehension | 1.90 | [-4.39, 8.20] | 0.59 | 104.53 | .555 |
| Trial Number × Condition | -0.18 | [-0.39, 0.02] | -1.77 | 3303.08 | .076. |
| Trial Number × WIAT-II Comprehension | 0.07 | [-0.04, 0.18] | 1.21 | 3300.95 | .226 |
| Condition × WIAT-II Comprehension | -0.18 | [-5.14, 4.78] | -0.07 | 351.61 | .943 |
| Trial Number × Condition × WIAT-II Comprehension | -0.06 | [-0.29, 0.17] | -0.51 | 3300.98 | .611 |

The baseline of the condition term is lower comprehension demands. Estimates represent the change when going from lower to higher comprehension demands.

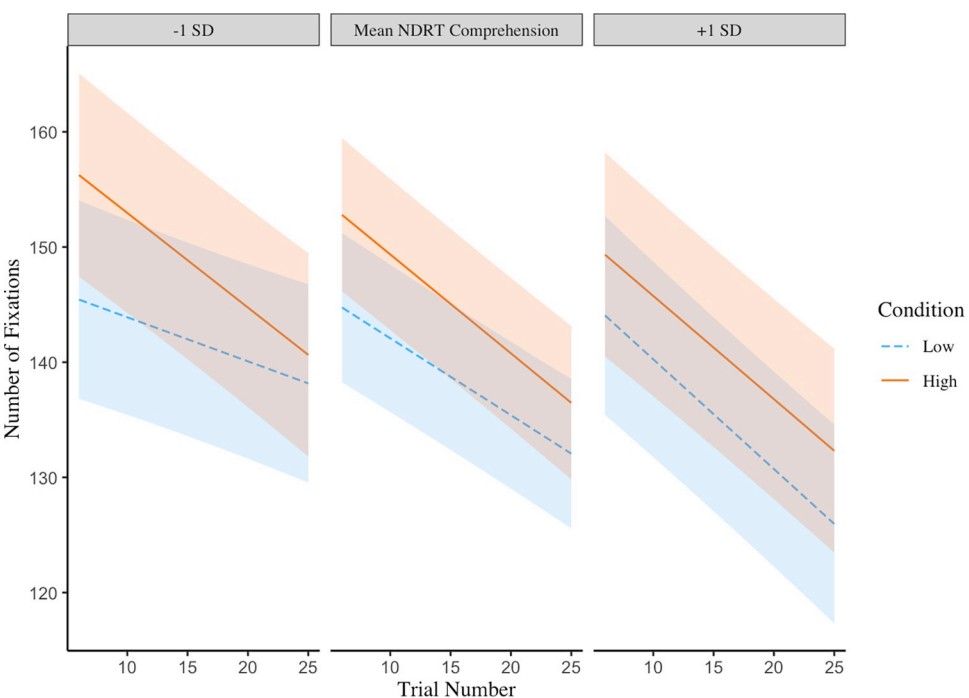

**Fig 1. A three-way interaction between NDRT comprehension scores, condition (higher vs lower comprehension demands) and trial number on the number of fixations made when reading a paragraph.** Shaded areas represent 95% confidence intervals.

comprehension demands were low, this decrease was not as steep. This pattern may indicate that less skilled comprehenders were nearing floor effects where they were close to the minimum number of fixations that they could accommodate whilst still reading for comprehension when comprehension demands were low.

No significant interactions or individual differences were observed for scores on the WIAT-II comprehension test, and in this model a trial by condition interaction was marginally significant (Table 3).

**Average fixation duration.** Tables 4 and 5 present models for average fixation durations. These models indicated that overall, average fixation durations increased slightly over trials. A three-way interaction between trial number, condition and scores on the NDRT comprehension test was observed (Table 4). Fig 2 shows that for low scorers on the NDRT average fixation durations increased from early to late trials in the experiment. For the high scorers a different pattern emerges depending on the comprehension demands with average fixation time going up when comprehension demands are high and going down when they are low.

WIAT-II comprehension scores were not significant predictors of average fixation durations (Table 5), though an interaction of trial by condition was marginally significant.

**Average forward saccade length.** Models for average forward saccade lengths are displayed in Tables 6 and 7. In both models, longer forward saccades were observed for passages with lower comprehension demands than for identical passages with higher comprehension demands. A slight increase in forward saccade length over trials was also predicted by both models. Table 6 shows that a three-way interaction between trials, conditions and NDRT comprehension scores was significant though numerically small. Differences can be seen in Fig 3 where those who scored highly on the NDRT made slightly longer forward saccades when comprehension demands were low compared to when comprehension demands were high,

**Table 4. LMM for average fixation durations predicted by NDRT comprehension and interactions with trial number and condition.**

| | β | 95% CI | t | df | p |
|---|---|---|---|---|---|
| Intercept | 205.48 | [200.82, 210.14] | 86.42 | 98.98 | < .001 *** |
| Trial Number | 0.10 | [0.05, 0.15] | 4.00 | 3300.34 | < .001 *** |
| Condition | -0.41 | [-2.71, 1.89] | -0.35 | 280.96 | .729 |
| NDRT Comprehension | -2.18 | [-6.75, 2.38] | -0.94 | 94.13 | .351 |
| Trial Number × Condition | 0.07 | [-0.03, 0.17] | 1.43 | 3301.82 | .154 |
| Trial Number × NDRT Comprehension | -0.07 | [-0.12, -0.03] | -2.97 | 3299.18 | .003 ** |
| Condition × NDRT Comprehension | -2.62 | [-4.89, -0.34] | -2.26 | 278.34 | .025 * |
| Trial Number × Condition × NDRT Comprehension | 0.12 | [0.02, 0.21] | 2.31 | 3296.20 | .021 * |

The baseline of the condition term is lower comprehension demands. Estimates represent the change when going from lower to higher comprehension demands.

but in both comprehension demand conditions forward saccade lengths became longer further in the experiment. Low scorers also made longer forward saccades when comprehension demands were low than when they were high, however the average length of their forward saccades only increased over time when comprehension demands were high. When comprehension demands were low, these readers did not make longer forward saccades over trials, with comparable average forward saccade lengths across all trials.

No significant effects of individual differences in WIAT-II comprehension test scores were observed for average forward saccade lengths (Table 7).

**Total passage reading times.** Models for total passage reading times are presented in Tables 8 and 9. In both models, passages in trials that occurred later in the experiment for both conditions were read more quickly than earlier passages. Passages were also read more quickly when comprehension demands were low compared to when comprehension demands were high (this was significant in both models). The model presented in Table 8 also revealed that total passage reading times were influenced by a three-way interaction between trials, conditions and NDRT comprehension scores. Fig 4 shows that high scorers consistently read passages more quickly towards the end of the experimental conditions than at the beginning, and read passages with lower comprehension demands more quickly than passages with higher comprehension demands. High scorers also read more quickly than low scorers by the end of the experiment in both conditions.

**Table 5. LMM for average fixation durations predicted by WIAT-II comprehension and interactions with trial number and condition.**

| | β | 95% CI | t | df | p |
|---|---|---|---|---|---|
| Intercept | 205.58 | [200.90, 210.26] | 86.08 | 98.93 | < .001 *** |
| Trial Number | 0.09 | [0.04, 0.14] | 3.64 | 3297.93 | < .001 *** |
| Condition | -0.65 | [-2.96, 1.66] | -0.55 | 280.78 | .580 |
| WIAT-II Comprehension | -3.27 | [-8.44, 1.90] | -1.24 | 94.18 | .218 |
| Trial Number × Condition | 0.09 | [-0.01, 0.19] | 1.75 | 3298.81 | .080. |
| Trial Number × WIAT-II Comprehension | 0.01 | [-0.04, 0.07] | 0.38 | 3297.93 | .703 |
| Condition × WIAT-II Comprehension | -0.02 | [-2.60, 2.56] | -0.01 | 278.01 | .989 |
| Trial Number × Condition × WIAT-II Comprehension | -0.05 | [-0.16, 0.06] | -0.87 | 3298.43 | .383 |

The baseline of the condition term is lower comprehension demands. Estimates represent the change when going from lower to higher comprehension demands.

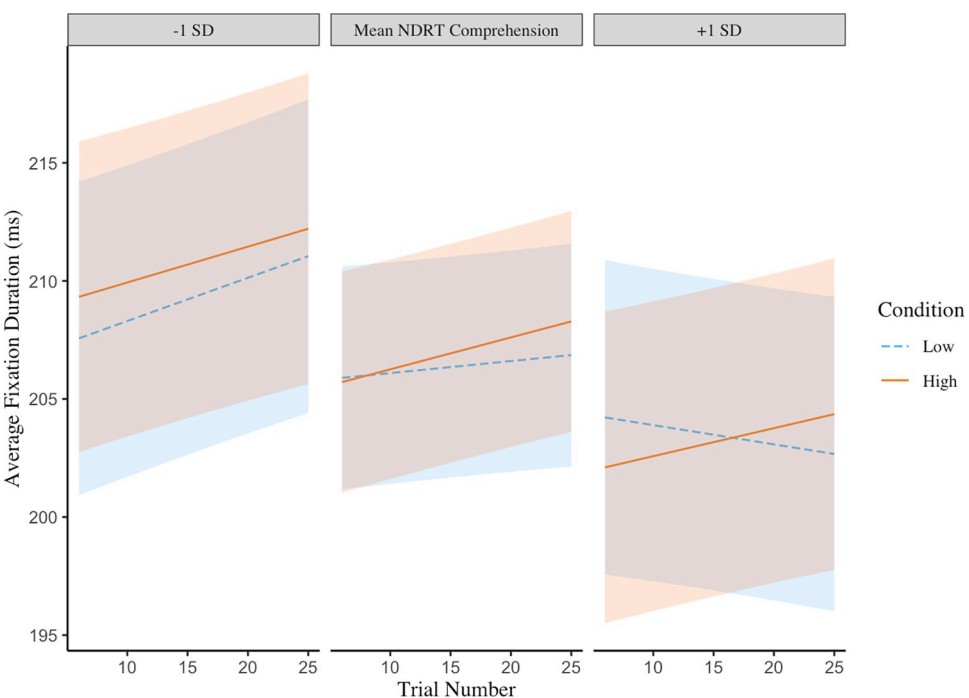

**Fig 2. A three-way interaction between NDRT comprehension scores, condition (higher vs lower comprehension demands) and trial number on average fixation durations when reading a paragraph.** Shaded areas represent 95% confidence intervals.

Low scorers on the NDRT comprehension scores displayed a different pattern, where their reading times were longer when comprehension demands were high compared to low at the beginning of the experiment and decreased over trials. However, when comprehension demands were low a potential floor effect was observed for low scorers where only a small decrease in reading times across trials was seen for passages.

No other significant effects were observed in the model including the WAIT-II comprehension scores (Table 9).

**Accuracy.** Neither model showed significant differences in accuracy for high (M = 84%, SD = 10.69) compared to lower comprehension demands (M = 73.78%, SD = 7.65), or across

**Table 6. LMM for average forward saccade length predicted by NDRT comprehension and interactions with trial number and condition.**

|  | β | 95% CI | t | df | p |
|---|---|---|---|---|---|
| Intercept | 5.89 | [5.70, 6.08] | 60.79 | 104.80 | < .001 *** |
| Trial Number | 0.01 | [0.01, 0.01] | 7.37 | 3279.81 | < .001 *** |
| Condition | -0.14 | [-0.24, -0.04] | -2.72 | 340.04 | .007 ** |
| NDRT Comprehension | 0.02 | [-0.16, 0.21] | 0.26 | 96.07 | .797 |
| Trial Number × Condition | 0.00 | [0.00, 0.01] | 1.68 | 3289.38 | .092. |
| Trial Number × NDRT Comprehension | 0.01 | [0.00, 0.01] | 5.19 | 2762.05 | < .001 *** |
| Condition × NDRT Comprehension | 0.10 | [0.00, 0.20] | 1.94 | 334.43 | .054. |
| Trial Number × Condition × NDRT Comprehension | -0.01 | [-0.01, 0.00] | -2.16 | 2633.99 | .031 * |

Note. The baseline of the condition term is lower comprehension demands. Estimates represent the change when going from lower to higher comprehension demands.

**Table 7. LMM for average forward saccade length predicted by WIAT-II comprehension and interactions with trial number and condition.**

|  | β | 95% CI | t | df | p |
|---|---|---|---|---|---|
| Intercept | 5.88 | [5.69, 6.07] | 60.24 | 104.60 | < .001 *** |
| Trial Number | 0.01 | [0.01, 0.01] | 7.99 | 3284.76 | < .001 *** |
| Condition | -0.14 | [-0.24, -0.04] | -2.65 | 351.51 | .009 ** |
| WIAT-II Comprehension | 0.06 | [-0.15, 0.27] | 0.54 | 96.75 | .591 |
| Trial Number × Condition | 0.00 | [0.00, 0.01] | 1.41 | 3287.86 | .159 |
| Trial Number × WIAT-II Comprehension | 0.00 | [0.00, 0.00] | -0.55 | 2314.64 | .580 |
| Condition × WIAT-II Comprehension | 0.01 | [-0.10, 0.12] | 0.19 | 353.67 | .851 |
| Trial Number × Condition × WIAT-II Comprehension | 0.00 | [0.00, 0.01] | 1.46 | 2661.39 | .144 |

Note. The baseline of the condition term is lower comprehension demands. Estimates represent the change when going from lower to higher comprehension demands.

trials (Tables 10 and 11). In terms of individual differences in accuracy, one interaction between trials, conditions and WIAT-II comprehension scores was found to be marginally significant (Table 11). The pattern observed suggested that when comprehension demands were high, high scorers on this test became more accurate over time, whereas low scorers became less accurate in later trials. However, these trends were marginal.

## Discussion

The current study investigated two offline reading comprehension tests (the NDRT and the WIAT-II) as predictors of individual differences in skilled readers' eye movements during

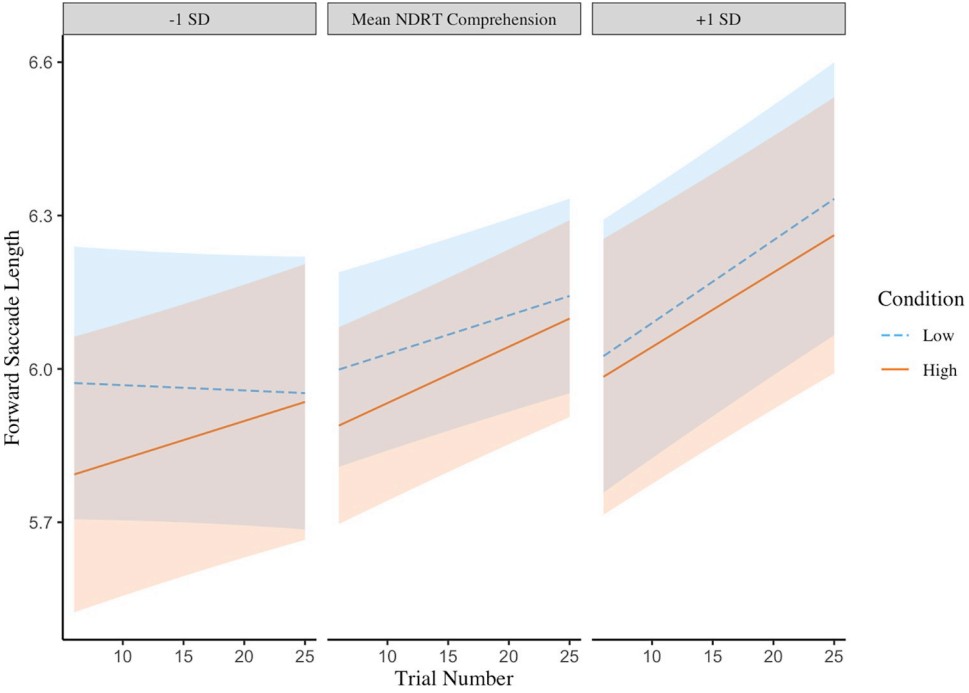

**Fig 3. A three-way interaction between NDRT comprehension scores, condition (higher vs lower comprehension demands) and trial number on the average forward saccade length when reading a paragraph.** Shaded areas represent 95% confidence intervals.

**Table 8. LMM for total passage reading times predicted by NDRT comprehension and interactions with trial number and condition.**

| | β | 95% CI | t | df | p |
|---|---|---|---|---|---|
| Intercept | 31996.48 | [30331.38, 33661.57] | 37.66 | 132.60 | < .001 *** |
| Trial Number | -144.55 | [-168.41, -120.70] | -11.88 | 3321.32 | < .001 *** |
| Condition | 2152.00 | [1081.86, 3222.15] | 3.94 | 304.94 | < .001 *** |
| NDRT Comprehension | -727.71 | [-2219.17, 763.74] | -0.96 | 100.48 | .341 |
| Trial Number × Condition | -42.15 | [-89.90, 5.59] | -1.73 | 3323.40 | .084. |
| Trial Number × NDRT Comprehension | -39.11 | [-62.68, -15.55] | -3.25 | 3320.14 | .001 ** |
| Condition × NDRT Comprehension | -1270.52 | [-2324.79, -216.25] | -2.36 | 311.73 | .019 |
| Trial Number × Condition × NDRT Comprehension | 66.07 | [19.03, 113.10] | 2.75 | 3321.15 | .006 ** |

The baseline of the condition term is lower comprehension demands. Estimates represent the change when going from lower to higher comprehension demands.

paragraph reading. Eye movement patterns were investigated under higher and lower comprehension demands and across trials. Parallel sets of analyses were conducted for each test to determine whether individual differences in offline comprehension tests predicted patterns in eye movement behaviour that was reflective of changes in comprehension demands, and whether readers adapted to comprehension demands over time. The main aim was to determine whether discrepancies arose between the two tests that claim to measure reading comprehension [10, 11], and a secondary aim was to investigate whether individual differences could be observed in the way that skilled readers adapted their reading strategies over time and in response to comprehension demands. First, we will focus on the overall patterns in the data across global eye movement measures, then on individual differences that were observed, and finally, we will discuss the two offline comprehension tests and differences in the predictive power associated with them for skilled readers.

**Table 9. LMM for total passage reading times predicted by WIAT-II comprehension and interactions with trial number and condition.**

| | β | 95% CI | t | df | p |
|---|---|---|---|---|---|
| Intercept | 31935.52 | [30246.02, 33625.02] | 37.05 | 131.55 | < .001 *** |
| Trial Number | -150.52 | [-174.47, -126.58] | -12.32 | 3321.70 | < .001 *** |
| Condition | 2059.34 | [986.54, 3132.14] | 3.76 | 308.20 | < .001 *** |
| WIAT-II Comprehension | -70.65 | [-1782.12, 1640.82] | -0.08 | 100.25 | .936 |
| Trial Number × Condition | -34.52 | [-82.43, 13.40] | -1.41 | 3322.80 | .158 |
| Trial Number × WIAT-II Comprehension | 18.14 | [-8.47, 44.76] | 1.34 | 3320.94 | .182 |
| Condition × WIAT-II Comprehension | -291.17 | [-1483.74, 901.41] | -0.48 | 314.56 | .633 |
| Trial Number × Condition × WIAT-II Comprehension | -9.23 | [-62.48, 44.03] | -0.34 | 3321.66 | .734 |

The baseline of the condition term is lower comprehension demands. Estimates represent the change when going from lower to higher comprehension demands.

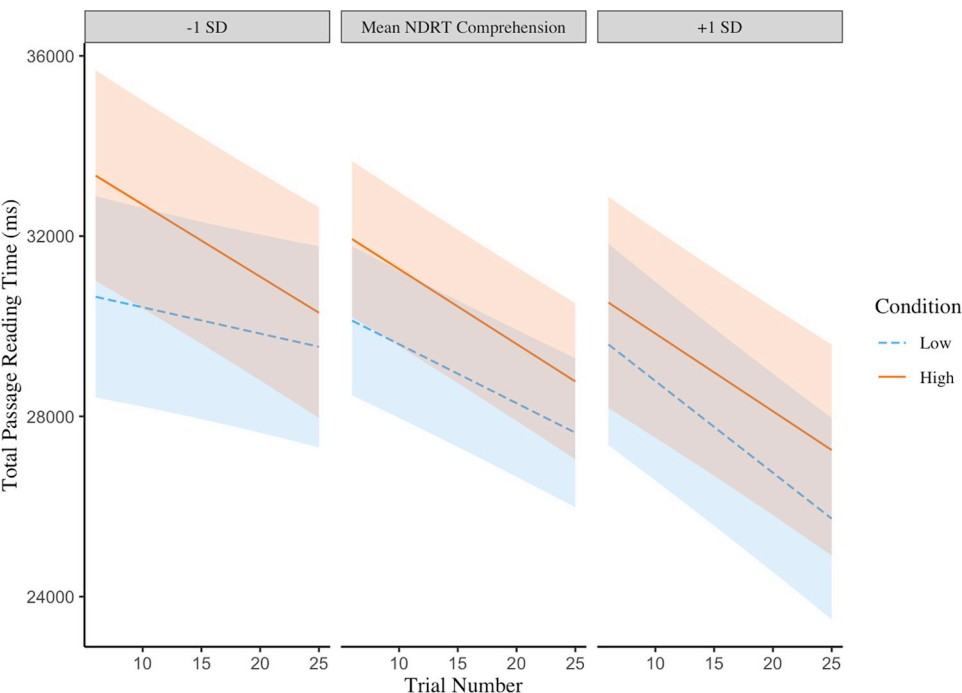

**Fig 4. A three-way interaction between NDRT comprehension scores, condition (higher vs lower comprehension demands) and trial number on total passage reading times.** Shaded areas represent 95% confidence intervals.

## Overall patterns

Overall, within an experimental block, paragraphs in later trials were read more quickly than in earlier trials. Reading strategies appeared to become more efficient, or perhaps more 'risky' [35] over time, with fewer fixations and increasing saccade lengths. Future investigations would need to include analyses of regressions to determine whether readers do use a riskier reading strategy in later trials since this may be more clearly observed though rereading behaviours. Participants were not more accurate on the comprehension questions in any one condition, or over time in the experiment. Though the increased difficulty in the higher comprehension demands condition was confirmed by a pre-test, it may be that for our skilled

**Table 10. Binomial GLMM for accuracy predicted by NDRT comprehension and interactions with trial number and condition.**

|  | β | 95% CI | z | p |
|---|---|---|---|---|
| Intercept | 2.65 | [1.54, 3.76] | 4.66 | < .001 *** |
| Trial Number | 0.01 | [-0.02, 0.03] | 0.56 | .576 |
| Condition | -1.57 | [-3.97, 0.84] | -1.28 | .201 |
| NDRT Comprehension | 0.05 | [-0.40, 0.49] | 0.20 | .841 |
| Trial Number × Condition | 0.02 | [-0.03, 0.07] | 0.66 | .510 |
| Trial Number × NDRT Comprehension | 0.01 | [-0.01, 0.04] | 0.96 | .338 |
| Condition × NDRT Comprehension | -0.09 | [-0.95, 0.76] | -0.21 | .832 |
| Trial Number × Condition × NDRT Comprehension | 0.00 | [-0.06, 0.05] | -0.16 | .873 |

The baseline of the condition term is lower comprehension demands. Estimates represent the change when going from lower to higher comprehension demands.

**Table 11. Binomial GLMM for accuracy predicted by WIAT-II comprehension and interactions with trial number and condition.**

| | β | 95% CI | z | p |
|---|---|---|---|---|
| Intercept | 2.58 | [1.49, 3.67] | 4.63 | < .001 *** |
| Trial Number | 0.01 | [-0.02, 0.04] | 0.79 | .430 |
| Condition | -1.48 | [-3.85, 0.88] | -1.23 | .220 |
| WIAT-II Comprehension | 0.06 | [-0.42, 0.53] | 0.23 | .815 |
| Trial Number × Condition | 0.02 | [-0.04, 0.07] | 0.59 | .558 |
| Trial Number × WIAT-II Comprehension | 0.01 | [-0.02, 0.04] | 0.84 | .401 |
| Condition × WIAT-II Comprehension | -0.59 | [-1.51, 0.33] | -1.26 | .209 |
| Trial Number × Condition × WIAT-II Comprehension | 0.05 | [-0.01, 0.11] | 1.69 | .090 . |

The baseline of the condition term is lower comprehension demands. Estimates represent the change when going from lower to higher comprehension demands.

readers, the higher demands were not enough to reduce their accuracy. Indeed, the pattern observed in the means suggested that participants had higher levels of accuracy when comprehension demands were high, which would be compatible with previous observations by Andrews and Veldre [44]. However, this difference was not significant in the analyses. We did, however, observe differences in eye movement patterns in response to the higher and lower comprehension demands. Readers were able to adjust their reading behaviours to the comprehension demands [36] and were able to read more thoroughly when comprehension demands were high and more superficially when comprehension demands were low [37]. Passages with higher comprehension demands were read more slowly, and featured more fixations and shorter saccades than passages with lower comprehension demands.

## Individual differences

Passage reading has the potential to introduce more variance in eye movement data compared to sentence reading simply due to the increase in processing demands, and the potential for allowing individual differences to be expressed in more varied ways. Slower reading and more rereading is often observed during passages compared to sentences [16]. Although our data do not echo Andrews and Veldre's [44] observations of shorter passage reading times, shorter average fixation durations and longer saccades directly related to individual differences in reading proficiency, their findings were based on a composite measure which included vocabulary, reading comprehension, reading rate and spelling, rather than comprehension alone. It may be that the direct effects of individual differences on fixation time measures observed by Andrews and Veldre [44] are better explained by other measures included in their composite score (e.g., spelling or vocabulary). In our analyses, individual differences as measured by off-line comprehension measures seem to predict the response to higher versus lower comprehension demands in the way that readers adapt over time.

Analysis of eye movements in relation to NDRT comprehension scores presented a clear picture of individual differences in response to comprehension demands. When reading behaviours were measured across trials, there were observable individual differences in the way that readers adapted their behaviour in response to comprehension demands. Differences between readers were smaller at the beginning of the experimental blocks and became larger in later trials where high scorers read more quickly, made fewer fixations and longer saccades than low scorers. High scorers read passages with higher comprehension demands more slowly, with more fixations and shorter saccades than passages with lower comprehension

demands, but the changes over time for higher and lower comprehension demands were comparable. In contrast, low scorers adapted their reading behaviours at a slower rate and approached a threshold for the fastest reading times, lowest number and duration of fixations, and the largest saccade lengths they were able to accommodate whilst reading for comprehension, even when comprehension demands were low. This evidence that less skilled comprehenders have a lower limit to how quickly they can read for comprehension than highly skilled comprehenders complements the general finding that less skilled readers often read more slowly and make longer fixations than more skilled readers [10, 11, 28, 29, 44].

## Offline comprehension measures

Critically, this pattern of results was highly dependent on which offline measure of comprehension was used to measure comprehension. Analyses of the same participants' eye movement data in relation to their scores on the WIAT-II comprehension test did not predict differences in eye movement patterns for different comprehension demands. Earlier, we described some differences in the format of each test that could indicate differences in the underlying skills measured by them. We return to these now to consider possible reasons why the NDRT revealed patterns in our data that the WIAT-II did not.

Higher comprehension accuracy is often observed for questions following narratives than expository texts [17]. Therefore, expository passages were selected for the current study to ensure that the materials were appropriate for skilled reading and to maximise the likelihood of finding variation in accuracy scores within this population. Potentially as a result of this choice, accuracy scores were not close to ceiling levels in the current study. The NDRT includes expository texts that are more similar to the current study materials than the WIAT-II comprehension test. Therefore, it is reasonable to suggest that comprehension based on similar test materials will account for a comparatively larger proportion of variance in reading behaviour. It has also been suggested that the NDRT is closely related to general knowledge [20, 21]. If the NDRT comprehension measure is highly related to general knowledge, we would expect to see higher levels of comprehension accuracy on our experimental questions for participants who score highly on the NDRT, but this was not observed.

In contrast, since the WIAT-II comprehension test includes some items that must be read aloud, it may feature some overlap with working memory processes [22]. However, our previous investigations of eye movement behaviours in sentence reading included the WIAT-II and a test of working memory (a backwards digit span task) amongst other reading skill predictors [10, 11]. These investigations did not suggest that there was much overlap between working memory and the WIAT-II comprehension test as they did not load together in principal components analyses [10, 11]. We highlighted some aspects of the WIAT-II comprehension subtest that may mean it also has less power to discriminate between skilled adult readers than the NDRT. First, narrative test comprehension is often higher than expository texts, which may indicate that portions of the WIAT-II comprehension test are not difficult enough to allow much variance within skilled readers. We also noted in the introduction that the reading aloud parts of the test may not be as informative about individual differences in adults as it is for children since adults rely on reading aloud less often [25], though further research would be needed to confirm this. In addition, the face-to-face aspect of the WIAT-II may lead to noisier data for adults where participants might experience performance anxiety.

It is also important that we acknowledge the potential impact that the low reliability of the WIAT-II may have had on results in this study. Low reliability in the WIAT-II may be the underlying reason for a weak correlation with the NDRT and null findings when predicting eye movement measures. Future research will need to explore whether the low reliability of the

WIAT-II that we observed is due to the specific population of skilled readers our study examined. Regardless, we maintain that when used to predict individual differences in eye movement patterns in adult readers that researchers should be cautious when selecting an appropriate test to use.

## Limitations

As is very common for participant samples which are mostly based on Psychology Undergraduate students, the current sample from the University of Southampton featured a high proportion of females, which may limit the generalisation to male participants.

We also note that the NDRT was not administered with the standard time limit. A precedent has been set by Andrews et al. [51] for administering a shortened version of the NDRT for researchers examining individual differences in skilled readers' eye movement patterns since it increases the variance between skilled readers. This choice might limit the comparability to research that uses the NDRT with the standard time limit. However, since our focus was on skilled readers and the NDRT shortened time limit is increasing in popularity in the field of examining skilled reading, we feel the choice for the shortened version of the NDRT was justified. In addition, reliability estimates for both comprehension tests based on our data were somewhat lower than the estimates given by each test manual [12, 13], therefore we note that these tests may have comparatively reduced reliability for university level populations.

All questions in the experimental conditions had two options from which participants were required to select an answer. Such limited response options may have limited the capacity to find differences in accuracy in our data. However, we note that this would not limit findings drawn from the eye movement record.

## Conclusion

Overall, it appears that the NDRT comprehension test (notably when following a half-timed procedure) is more sensitive to differences in eye movement behaviours in response to higher and lower comprehension demands observed between skilled adult readers compared to the WIAT-II comprehension test. Individual differences captured by the half-timed version of the NDRT have been previously shown to be sensitive to individual differences in skilled readers eye movements [50]. The current study extends this and suggests it can be used to predict differences in eye movement behaviours across trials in response to varying comprehension demands. We highlight the importance of careful test selection when measuring eye movement behaviour in skilled adult readers and advise that comprehension tests should not be used interchangeably, because they *jingle* [14] and that researchers should exercise caution when selecting a reading comprehension test for future research. We echo advice from Flake and Fried [15] who call for transparency when reporting test selection processes and urge researchers to select comprehension tests that are clearly based on the theoretical concepts that the researcher wishes to assess.

## Acknowledgments

Special thanks to Karolina Vakulya for her work on data collection for this research.

## Author Contributions

**Conceptualization:** Denis Drieghe.

**Data curation:** Charlotte E. Lee, Denis Drieghe.

**Formal analysis:** Charlotte E. Lee, Denis Drieghe.

**Funding acquisition:** Charlotte E. Lee, Denis Drieghe.

**Investigation:** Charlotte E. Lee, Denis Drieghe.

**Methodology:** Charlotte E. Lee, Denis Drieghe.

**Project administration:** Charlotte E. Lee, Denis Drieghe.

**Resources:** Charlotte E. Lee, Denis Drieghe.

**Software:** Charlotte E. Lee, Denis Drieghe.

**Supervision:** Denis Drieghe.

**Validation:** Charlotte E. Lee, Denis Drieghe.

**Visualization:** Charlotte E. Lee, Denis Drieghe.

**Writing – original draft:** Charlotte E. Lee, Hayward J. Godwin, Denis Drieghe.

**Writing – review & editing:** Charlotte E. Lee, Hayward J. Godwin, Denis Drieghe.

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
