## [Decision Letter · Decision Letter 0]

4 Sep 2023

PONE-D-23-23595The jingle fallacy in comprehension tests for readingPLOS ONE

Dear Dr. Lee,

Thank you for submitting your manuscript to PLOS ONE. After careful consideration, we feel that it has merit but does not fully meet PLOS ONE’s publication criteria as it currently stands. Therefore, we invite you to submit a revised version of the manuscript that addresses the points raised during the review process.

 Please submit your revised manuscript by Oct 19 2023 11:59PM. If you will need more time than this to complete your revisions, please reply to this message or contact the journal office at plosone@plos.org. Please include the following items when submitting your revised manuscript:A rebuttal letter that responds to each point raised by the academic editor and reviewer(s). You should upload this letter as a separate file labeled 'Response to Reviewers'.A marked-up copy of your manuscript that highlights changes made to the original version. You should upload this as a separate file labeled 'Revised Manuscript with Track Changes'.An unmarked version of your revised paper without tracked changes. You should upload this as a separate file labeled 'Manuscript'.

We look forward to receiving your revised manuscript.

Kind regards,

Michael Flor

Academic Editor

PLOS ONE

Journal Requirements:

Additional Editor Comments:

Please thoroughly consider the comments from the reviewers. Please also consider the following points: 1.Lines 285-287:About the 9 participants that were removed "due poor overall accuracy" - accuracy on what? the reading tests? the experimental data? 2.Lines 321-323.The high- and low- comprehension demand questions for the experiment had different averages, 23.71 vs. 19.97.

Although the difference is statistically significant, the  difference is less than 4 points on a 100-point scale and very close to the lower (easier) part. It is not quite convincing that the high-comprehension-demand questions were really more difficult. At least some discussion of this is needed. 3.Lines 370-376: the correlation of NDRT with WIAT-II was just r=0.22.Doesn't the low correlation (over same subjects) indicate that the test are not measuring the same thing?Some discussion around this is needed. Are you aware of any other studies that correlated results on those tests?If the tests are indeed uncorrelated or low-correlated, the 'jingle fallacy' would be shown without the need for eye-tracking studies. 4.The notion of high and low scorers is used (line 417), but there is no description on how participants were assigned to such groups for each reading test.  The high- low- scorers implies a dichotomous split.

Consider presenting a scatter plot of WIAT vs NDRT scores - does the distribution of scores on each support a dichotomous split on each test - or maybe more than 2 groups should be considered for each? 5.Given the study by Coleman et al., cited  on lines 96-98, that  college students could answer the

 questions on NDRT comprehension tests and achieve a greater-than-chance level of accuracy

 without actually reading the associated passages - why do you still recommend NDRT as a reading test - rather than a general knowledge test?  Some discussion is needed. 6.Regarding the study of Eason et al., cited on lines 98-102.First, you should note that they studied a different test (not NDRT). How that relates to 'cognitive skills' that might underly WIAT  or NDRT is unclear.Moreover, your own citing mentions "when texts and questions were complex". But the text and questions in your study were not complex (see point 3 above). So how does the Eason study  fit here?  ==============================

Reviewers' comments:

Reviewer's Responses to Questions

**Comments to the Author**

1. Is the manuscript technically sound, and do the data support the conclusions?

Reviewer #1: Partly

Reviewer #2: Partly

2. Has the statistical analysis been performed appropriately and rigorously? 

Reviewer #1: Yes

Reviewer #2: Yes

3. Have the authors made all data underlying the findings in their manuscript fully available?

Reviewer #1: Yes

Reviewer #2: Yes

4. Is the manuscript presented in an intelligible fashion and written in standard English?

Reviewer #1: Yes

Reviewer #2: Yes

5. Review Comments to the Author

Reviewer #1: This study examines the influence of comprehension demands on eye movements as a function of the type of comprehension test. The results suggest that students are generally sensitive to task demands, but there were key differences between skilled and less skilled readers suggesting the existence of a threshold (for less skilled readers). There were also different patterns across the comprehension measures, suggesting they are not interchangeable. Overall, this result is largely consistent with prior research (e.g., Cutting & Scarborough, 2006) that suggests different comprehension measures may tap differential aspects of the construct.

In short, the paper is well written and I commend the authors for pointing out key differences in the two comprehensions tests upfront. Differences in length, genre, administration and production makes these tests more distinct than many other possible comprehension test comparisons. In addition, the review of the eye-tracking research related to this topic is well done and informative. Similarly, the hypotheses are clear and supported by prior research.

That said, the main cautionary note for this study concerns any major claims made between the two comprehension measures (e.g., NDRT is easier or harder than the WIAT, or the low correlation between the two). These are not warranted as the NDRT was not administered with adequate time limits. Please note the non-standardized administration of the NDRT as a limitation.

Larger issues

The terms low and high comprehension demands are somewhat misleading. I was expecting deeper comprehension questions in the high demand condition. Maybe it would be more accurate to call the conditions “lower” and “higher” as the judgement is relative, not absolute. Similarly, it is not that the questions are deeper per se, but the length and detail of the response required that makes it harder. In other words, “what the passage is about” and “what is the main idea” are not that different in terms of the question alone.

What were the properties of the comprehension questions from the free online practice tests? The limited response options may impact the reliability and subsequently the conclusions.

Pg 14 I was surprised you didn’t ask students to rate the perceived complexity of the paragraph rather than ask about its naturalness (maybe a suggestion for future research). To this point, there doesn’t seem to be any analyses on the naturalness dimension besides the mean. How does one rate how natural passage is? Relative to what criterion? Seems odd.

Pg16 Why was the NDRT stopped after 10 minutes? This is does not represent standardized conditions and thus may impact the interpretations across tests. This should be mentioned as a limitation in the limitations section.

Maybe one reason the NDRT and the WIAT were not correlated very high was the fact that the NDRT was speeded because of the reduced administration time. In general, many comprehension assessments correlate between .60-.80. .22 seems very low.

Pg19 “Fig 1 shows that high scorers on the NDRT comprehension test reduced the number of fixations further into the experimental session”. Were students aware of how much time they had? If so, they could have been rushing as they progressed.

Minor issues

The sample is very unbalanced in terms of gender. This should be mentioned in the limitations section.

Pg 13 it sounds like you programmed the NDRT on your own (Qualtrics). If the publisher reads the article, they may not be too happy if there was no permission to do so.

Pg14 for transparency, it would be helpful to at least name the other tasks given to participants in the experiment that were outside the scope of this paper as the type of task may relate to interference. The timings would also be helpful as it relates to issues such as fatigue.

Pg15 how are the literal and inferential questions related to the low and high demand tasks? Please provide some clarity.

Pg16 again, please name the unrelated tasks (other online tasks) and timings for the second session for transparency. Also what are the reliabilities of the comprehension tests in session 2.

Reviewer #2: I really enjoyed reading this well-written manuscript. I only have one question. What is the reliability of the WIAT-II comprehension test (and the NDRT comprehension) calculated from this sample? This is critical information, especially given that the two tests have low correlation. One interpretation is that the two tests are indeed measuring different constructs as the authors propose. However, this still is the possibility that the low correlation is a result of low reliability of one or two of the tests. Looking at the WIAT-II comprehension scores, M=110.47 and SD=9.37, if these are standardized scores, this sample is quite high performing on WIAT-II (the WIAT-II population M=100 and SD=15), with this sample's average score .67 SD above the population mean. There might be a ceiling effect on this test, which prevents its predictive power to be fully realized, which could explain the low correlation to NDRT and its non-significant relation to eye-tracking data.

I also briefly looked at the data file shared by the authors. Is the WIAT-II comprehension scores saved under the column named "WIATTotal"? If so, the data do not match what is reported in the manuscript, which reports that the range is 71-124 but in the data file the range is 84-134.

6. PLOS authors have the option to publish the peer review history of their article (what does this mean?). If published, this will include your full peer review and any attached files.

Reviewer #1: No

Reviewer #2: **Yes: **Zuowei Wang

---

## [Author Response · Author response to Decision Letter 0]

5 Oct 2023

Dear Dr Flor,

Before responding to the issues raised by yourself and the reviewers, I would like to thank you all on behalf of myself and my co-authors for the very helpful comments on the previous version of this paper. 

We believe we have now successfully addressed the issues raised by yourself and the reviewers. We highlighted the changes in the manuscript in yellow so you can quickly see what has changed.

Kind regards, 

Charlotte Lee

1.

Lines 285-287:

About the 9 participants that were removed "due poor overall accuracy"

 - accuracy on what? the reading tests? the experimental data?

Response: This refers to poor accuracy on the comprehension questions during the eye-tracking experiment, we have added this clarification on line 284. 

2.

Lines 321-323.

The high- and low- comprehension demand questions for the experiment had different averages, 23.71 vs. 19.97.

Although the difference is statistically significant, the difference is less than 4 points on a 100-point scale and very close to the lower (easier) part. It is not quite convincing that the high-comprehension-demand questions were really more difficult. At least some discussion of this is needed.

Response: This is a valid point and was raised by Reviewer 1 as well. We have now amended our description of the conditions throughout the manuscript to ‘higher and lower comprehension demands’ instead of high and low as suggested by Reviewer 1.

3.

Lines 370-376: the correlation of NDRT with WIAT-II was just r=0.22.

Doesn't the low correlation (over same subjects) indicate that the test are not measuring the same thing?

Some discussion around this is needed. Are you aware of any other studies that correlated results on those tests?

If the tests are indeed uncorrelated or low-correlated, the 'jingle fallacy' would be shown without the need for eye-tracking studies.

Response: We have shown in our previous work (Lee et al. submitted) that these tests were weakly correlated (mentioned in line 66) and are not aware of any other direct comparisons in the literature. Indeed, the low correlation between these tests suggests that the tests might not measure the same construct. However, the low correlation is uninformative regarding which of the two measures (or indeed if any) pick up comprehension difficulties (as reflected in eye movements).

With the addition of eye-tracking methodology we can further examine where these discrepancies lie – e.g., they may each measure a different aspect of comprehension seen in early or late eye movement measures. This issue is discussed in line 75 “The aim of the current paper …”.

4.

The notion of high and low scorers is used (line 417), but there is no description on how participants were assigned to such groups for each reading test. The high- low- scorers implies a dichotomous split.

Consider presenting a scatter plot of WIAT vs NDRT scores - does the distribution of scores on each support a dichotomous split on each test - or maybe more than 2 groups should be considered for each?

Response: Throughout all the analyses reported in the paper, the WIAT and NDRT scores were treated as continuous predictors. Our graphics aid the interpretation of the continuous effects by the data with separate panels for mean +/- 1SD for each reading test. We have now included a note about this on line 429.

5.

Given the study by Coleman et al., cited on lines 96-98, that college students could answer the

 questions on NDRT comprehension tests and achieve a greater-than-chance level of accuracy

 without actually reading the associated passages - why do you still recommend NDRT as a reading test - rather than a general knowledge test? Some discussion is needed.

Response: This is again a very good point and we are happy to add a clarification. Greater than chance accuracy was achieved by participants in the Coleman et al study when responding to questions without actually reading the passages but accuracy was at 44-47% whereas chance level was at 20%. Clearly the scores were well below ceiling level and the test is testing more than just general knowledge. We clarified this on line 98.

6.

Regarding the study of Eason et al., cited on lines 98-102.

First, you should note that they studied a different test (not NDRT). How that relates to 'cognitive skills' that might underly WIAT or NDRT is unclear.

Moreover, your own citing mentions "when texts and questions were complex". But the text and questions in your study were not complex (see point 3 above). So how does the Eason study fit here? 

Response: We see how this reference is adding complexity without adding substantial insights. We have removed the reference. 

Reviewer #1: This study examines the influence of comprehension demands on eye movements as a function of the type of comprehension test. The results suggest that students are generally sensitive to task demands, but there were key differences between skilled and less skilled readers suggesting the existence of a threshold (for less skilled readers). There were also different patterns across the comprehension measures, suggesting they are not interchangeable. Overall, this result is largely consistent with prior research (e.g., Cutting & Scarborough, 2006) that suggests different comprehension measures may tap differential aspects of the construct.

In short, the paper is well written and I commend the authors for pointing out key differences in the two comprehensions tests upfront. Differences in length, genre, administration and production makes these tests more distinct than many other possible comprehension test comparisons. In addition, the review of the eye-tracking research related to this topic is well done and informative. Similarly, the hypotheses are clear and supported by prior research.

That said, the main cautionary note for this study concerns any major claims made between the two comprehension measures (e.g., NDRT is easier or harder than the WIAT, or the low correlation between the two). These are not warranted as the NDRT was not administered with adequate time limits. Please note the non-standardized administration of the NDRT as a limitation.

Response: We thank the reviewer for their very positive review of our paper. 

Our choice for a non-standardized administration of the NDRT came with a cost being the reduced comparability with experiments using the standardized administration. However, even though we acknowledge the cost of this choice, we are confident in our choice for the reduced time limit because of a) the demonstrated increased variance within the population which we study, i.e., skilled readers and b) the fact that the shortened version is being used more and more for this group of participants.

We write on lines 662 – 669: “We also note that the NDRT was not administered with the standard time limit. A precedent has been set by Andrews et al [51] for administering a shortened version of the NDRT for researchers examining individual differences in skilled readers’ eye movement patterns since it increases the variance between skilled readers. This choice might limit the comparability to research that uses the NDRT with the standard time limit. However, since our focus was on skilled readers and the NDRT shortened time limit is increasing in popularity in the field of examining skilled reading, we feel the choice for the shortened version of the NDRT was justified.“

Larger issues

The terms low and high comprehension demands are somewhat misleading. I was expecting deeper comprehension questions in the high demand condition. Maybe it would be more accurate to call the conditions “lower” and “higher” as the judgement is relative, not absolute. Similarly, it is not that the questions are deeper per se, but the length and detail of the response required that makes it harder. In other words, “what the passage is about” and “what is the main idea” are not that different in terms of the question alone.

Response: We have now amended this description of the comprehension demands throughout the manuscript to reflect ‘higher’ and ‘lower’ comprehension demands as suggested by the reviewer. And we also added a clarification of the manipulation on line 315.

What were the properties of the comprehension questions from the free online practice tests? The limited response options may impact the reliability and subsequently the conclusions.

Response: We have included more description of how the questions were adapted from online tests in line 315-319. Original questions were the ‘higher demands’ questions, and we created a ‘lower demands alternative for each of them. All questions had two options from which participants were required to select an answer. We may have limited capacity to find differences in accuracy due to the limited response options and have now included this as a limitation in line 670. However, this would not limit findings drawn from the eye movement record.

Pg 14 I was surprised you didn’t ask students to rate the perceived complexity of the paragraph rather than ask about its naturalness (maybe a suggestion for future research). To this point, there doesn’t seem to be any analyses on the naturalness dimension besides the mean. How does one rate how natural passage is? Relative to what criterion? Seems odd.

Response: The purpose of this pre-screening was exclusively to identify any passages that were outliers in terms of how readable the material was. We have added clarification of this on line 321. Since the exact same texts were presented in both experimental conditions, the materials read by participants did not differ in complexity between conditions. Also, for our analyses, the passages were included in the random effects structure to ensure that findings were not dependent on which text was presented. 

Pg16 Why was the NDRT stopped after 10 minutes? This is does not represent standardized conditions and thus may impact the interpretations across tests. This should be mentioned as a limitation in the limitations section. Maybe one reason the NDRT and the WIAT were not correlated very high was the fact that the NDRT was speeded because of the reduced administration time. In general, many comprehension assessments correlate between .60-.80. .22 seems very low. Pg19 “Fig 1 shows that high scorers on the NDRT comprehension test reduced the number of fixations further into the experimental session”. Were students aware of how much time they had? If so, they could have been rushing as they progressed.

Response: We addressed the issue of our choice for the reduced administration time for the NDRT when replying to the comments from Reviewer 1. The limitation is addressed in the discussion as is a justification for the choice of the reduced administration time. 

Participants were not aware of the time limit for the NDRT, so we have no reason to suggest that it promoted speeded reading. We now state this explicitly on line 372. Also, the experimental data where we see a reduced number of fixations is for the texts presented in the eye tracking session where there were no time limits – not during the NDRT which is an “offline” (not eye-tracking) measure. 

Minor issues

The sample is very unbalanced in terms of gender. This should be mentioned in the limitations section.

Response: We have now noted this limitation on line 659 and also that this imbalance is very typical for psychology undergraduate students.

Pg 13 it sounds like you programmed the NDRT on your own (Qualtrics). If the publisher reads the article, they may not be too happy if there was no permission to do so.

Response: Well spotted! We forgot to mention this. Paper copies of the NDRT were purchased from the publisher and voided whenever we ran a participant online. This information has now been included on line 300.

Pg14 for transparency, it would be helpful to at least name the other tasks given to participants in the experiment that were outside the scope of this paper as the type of task may relate to interference. The timings would also be helpful as it relates to issues such as fatigue.

Response: This information has now been included on lines 334 and 363.

Pg15 how are the literal and inferential questions related to the low and high demand tasks? Please provide some clarity.

Response: This passage on literal and inferential questions refers to the WIAT-II task which was used as an offline (not eye-tracking) measure of reading ability. The high and low comprehension demands were conditions in the eye tracking experimental session. So, they were not related.

Pg16 again, please name the unrelated tasks (other online tasks) and timings for the second session for transparency. Also what are the reliabilities of the comprehension tests in session 2.

Response: This information has now been included on lines 334 and 363.

Reviewer #2: 

I really enjoyed reading this well-written manuscript. I only have one question. What is the reliability of the WIAT-II comprehension test (and the NDRT comprehension) calculated from this sample? This is critical information, especially given that the two tests have low correlation. One interpretation is that the two tests are indeed measuring different constructs as the authors propose. However, this still is the possibility that the low correlation is a result of low reliability of one or two of the tests. 

Response: We thank the reviewer for their positive appreciation of our paper. 

On lines 375-378 we added:

“The reliability for the WIAT-II comprehension test was calculated using the split-half reliability with spearman-brown adjustment to be .76. Given the low number of items for the NDRT comprehension test, we calculated reliability using Cronbach’s alpha to be .80.”

Looking at the WIAT-II comprehension scores, M=110.47 and SD=9.37, if these are standardized scores, this sample is quite high performing on WIAT-II (the WIAT-II population M=100 and SD=15), with this sample's average score .67 SD above the population mean. There might be a ceiling effect on this test, which prevents its predictive power to be fully realized, which could explain the low correlation to NDRT and its non-significant relation to eye-tracking data.

Response: The scores used in this study are for the WIAT-II comprehension subtest only rather than the entire WIAT-II. The mean for the comprehension subtest in the general population is 103.11 with SD 16.73 according to adult scoring supplement. We also focussed on skilled readers for this experiment, so our sample is from average to very skilled readers, therefore an above average sample on the WIAT-II is fully expected. The notion that we focus on skilled readers is mentioned throughout the paper. 

I also briefly looked at the data file shared by the authors. Is the WIAT-II comprehension scores saved under the column named "WIATTotal"? If so, the data do not match what is reported in the manuscript, which reports that the range is 71-124 but in the data file the range is 84-134.

Response: The correct column in the datafile for the raw data is ComprehensionStandardScore – these are scores for the comprehension subtest only (the WIATTotal column includes the other reading subtests, we used this for data cleaning – for example removing participants who were below average readers).

---

## [Decision Letter · Decision Letter 1]

19 Dec 2023

PONE-D-23-23595R1The jingle fallacy in comprehension tests for readingPLOS ONE

Dear Dr. Lee,

Thank you for submitting your manuscript to PLOS ONE. After careful consideration, we feel that it has merit but does not fully meet PLOS ONE’s publication criteria as it currently stands. Therefore, we invite you to submit a revised version of the manuscript that addresses the points raised during the review process.

We look forward to receiving your revised manuscript.

Kind regards,

Vanessa Carels

Staff Editor

PLOS ONE

Reviewers' comments:

Reviewer's Responses to Questions

**Comments to the Author**

1. If the authors have adequately addressed your comments raised in a previous round of review and you feel that this manuscript is now acceptable for publication, you may indicate that here to bypass the “Comments to the Author” section, enter your conflict of interest statement in the “Confidential to Editor” section, and submit your "Accept" recommendation.

Reviewer #1: All comments have been addressed

Reviewer #2: (No Response)

2. Is the manuscript technically sound, and do the data support the conclusions?

Reviewer #1: Yes

Reviewer #2: Partly

3. Has the statistical analysis been performed appropriately and rigorously? 

Reviewer #1: Yes

Reviewer #2: No

4. Have the authors made all data underlying the findings in their manuscript fully available?

Reviewer #1: Yes

Reviewer #2: No

5. Is the manuscript presented in an intelligible fashion and written in standard English?

Reviewer #1: Yes

Reviewer #2: Yes

6. Review Comments to the Author

Reviewer #1: Thanks for the opportunity to review a revised version of the manuscript entitled “The jingle fallacy in comprehension tests for reading”. The authors have been responsive to the reviews and I have no additional concerns.

Reviewer #2: The key argument of this manuscript is that comprehension tests can be measuring different constructs. This argument has been supported by research focusing on the role of decoding and language comprehension (Keenan et al., 2008, https://doi.org/10.1080/10888430802132279) or the role of different types of knowledge (Wang et al., 2021, https://doi.org/10.1016/j.learninstruc.2021.101462), among other studies. In all these studies, researchers compared the differences of reading comprehension tests by comparing students' behaviors and by correlating these comprehension measures to other measures, and this manuscript applied a similar approach. However, another important piece of information provided by priror research comparing different comprehension tests is the reliability of tests that are compared. We can only compare whether two tests are measuring the same construct after we are certain that both tests are highly and similarly reliable. Two tests, even measuring the same construct, can appear to have zero correlation if one or both of them are unreliable.

In my previous review, I expressed concerns about the reliability of the two reading comprehension tests. I appreciate the authors providing more information. However, the new information provided seems incomplete, and it only makes me more concerned about altnernative explanations for the low correlation between the two comprehension tests. First, for WIAT-II comprehension, the authors only reported adjusted reliability "split-half reliability with spearman-brown adjustment to be .76". What is Cronbach's alpha? What is the reason for reporting "adjusted reliability" rather than the original empirical reliability? The Spearman-Brown formula helps predict the reliability of a test with a different test length. What is the hypothetical test length that is used to calculate this "adjusted" reliabiliy? Please report the unadjusted empirical reliability.

Second, whereas the WIAT-II used split-half reliability (adjusted), the NDRT used Cronbach's alpha. To my understanding, the NDRT test was timed, so that not all students finished all items. How was this dealt with when calculating Cronbach's alpha? What is the split half reliability of NDRT? The reliability information provided by the authors do not reassure me about the possibility that the low correlation of the two tests is mainly a result of low reliability of either or both tests. More transparency is needed.

Third, using the data provided by the author (thanks again for sharing the data), I found that students' comprehension scores on both tests significantly deviates from normal distribution. There is a lack of students in the middle of the distribution. Most students were near the top, and some students near the bottom. Deviation from normal distribution may distort the estimation of correlations. Furthermore, from the dataset, the correlation between NDRT comprehension and NDRT vocabulary subtests was also surprisingly low, with r = .16. This again makes me worried that the NDRT test might not be reliable for this sample. One would expect much higher correlation between the two subtests of NDRT.

If it is indeed the case that the two comprehension tests (NDRT, WIAT-II) have low reliability in this study sample, this paper can still make a contribution to the field with the eye-tracking data. Most of the analysis section simply treats student performance on either comprehension tests as a predictor, and the three way interaction effects are interesting and informative. If the authors only uses median split based on students' comprehension scores, the suboptimal reliability of these comprehension measures might be OK -- you don't need a super reliable test to differentiate two groups of people. However, if the question is to compare construct coverage of the two comprehension tests, the demand on the reliability of the tests is much higher, which I'm afraid the current dataset/sample don't afford.

7. PLOS authors have the option to publish the peer review history of their article (what does this mean?). If published, this will include your full peer review and any attached files.

Reviewer #1: No

Reviewer #2: No

---

## [Author Response · Author response to Decision Letter 1]

13 Feb 2024

Dear, Dr. Carels

Thank you for taking the time to review this manuscript. My co-authors and I have now addressed reviewers’ comments in the text below and changes are indicated in the manuscript.

Kind regards,

Charlotte Lee

Review Comments to the Author

Reviewer #1: Thanks for the opportunity to review a revised version of the manuscript entitled “The jingle fallacy in comprehension tests for reading”. The authors have been responsive to the reviews and I have no additional concerns.

We would first like to thank both reviewers for their helpful comments on this manuscript and previous versions.

Reviewer #2: The key argument of this manuscript is that comprehension tests can be measuring different constructs. This argument has been supported by research focusing on the role of decoding and language comprehension (Keenan et al., 2008, https://doi.org/10.1080/10888430802132279) or the role of different types of knowledge (Wang et al., 2021, https://doi.org/10.1016/j.learninstruc.2021.101462), among other studies. In all these studies, researchers compared the differences of reading comprehension tests by comparing students' behaviors and by correlating these comprehension measures to other measures, and this manuscript applied a similar approach. However, another important piece of information provided by priror research comparing different comprehension tests is the reliability of tests that are compared. We can only compare whether two tests are measuring the same construct after we are certain that both tests are highly and similarly reliable. Two tests, even measuring the same construct, can appear to have zero correlation if one or both of them are unreliable.

In my previous review, I expressed concerns about the reliability of the two reading comprehension tests. I appreciate the authors providing more information. However, the new information provided seems incomplete, and it only makes me more concerned about altnernative explanations for the low correlation between the two comprehension tests. First, for WIAT-II comprehension, the authors only reported adjusted reliability "split-half reliability with spearman-brown adjustment to be .76". What is Cronbach's alpha? What is the reason for reporting "adjusted reliability" rather than the original empirical reliability? The Spearman-Brown formula helps predict the reliability of a test with a different test length. What is the hypothetical test length that is used to calculate this "adjusted" reliabiliy? Please report the unadjusted empirical reliability.

Second, whereas the WIAT-II used split-half reliability (adjusted), the NDRT used Cronbach's alpha. To my understanding, the NDRT test was timed, so that not all students finished all items. How was this dealt with when calculating Cronbach's alpha? What is the split half reliability of NDRT? The reliability information provided by the authors do not reassure me about the possibility that the low correlation of the two tests is mainly a result of low reliability of either or both tests. More transparency is needed.

Response: In response to comments from Reviewer #2, we have reassessed our use of reliability measures in our analyses. The rationale for originally using two different measures for the comprehension tests in our paper was that these were the reliability estimates given in each respective test user manual. It was reasoned that these would be appropriate to calculate to compare reliability in our data with reliability in the norming samples. Following the helpful comments from reviewer #2, we note that by using the WIAT comprehension test solely on adults, the test no longer differs in length (as it would if children took the test) and therefore does not require the spearman-brown adjustment as suggested by the manual. We agree with reviewer #2 that Cronbach’s alpha is therefore more suitable and now report this measure in the manuscript for both the NDRT and the WIAT-II comprehension tests. Reliability for the NDRT comprehension was more difficult to calculate given that participants may complete a different number of items in the given time. We report a Cronbach’s alpha estimate in which missing data have been removed and note that for participants who answer fewer items in the given time, test reliability will be automatically lower. These estimates are discussed in the method section on lines 375 -382 and again in the limitations section on lines 673 -676. Cronbach’s Alpha is still acceptable for both tests (>.60) but does seem to be lower than the reliability values reported in the user manual. This discrepancy is mentioned and discussed in the paper. 

Third, using the data provided by the author (thanks again for sharing the data), I found that students' comprehension scores on both tests significantly deviates from normal distribution. There is a lack of students in the middle of the distribution. Most students were near the top, and some students near the bottom. Deviation from normal distribution may distort the estimation of correlations. Furthermore, from the dataset, the correlation between NDRT comprehension and NDRT vocabulary subtests was also surprisingly low, with r = .16. This again makes me worried that the NDRT test might not be reliable for this sample. One would expect much higher correlation between the two subtests of NDRT.

Response: Regarding deviation of the test data from a normal distribution, we compared the original parametric correlation of the two tests (Pearson’s r = 0.22) with non-parametric correlations (Spearman’s ρ = 0.23) and found them to be extremely comparable. We hope this indicates that any deviation from the normal distribution will not have led to a different conclusion.

---

## [Decision Letter · Decision Letter 2]

10 May 2024

PONE-D-23-23595R2The jingle fallacy in comprehension tests for readingPLOS ONE

Dear Dr. Lee,

Thank you for submitting your manuscript to PLOS ONE. After careful consideration, we feel that it has merit but does not fully meet PLOS ONE’s publication criteria as it currently stands. Therefore, we invite you to submit a revised version of the manuscript that addresses the points raised during the review process.

We look forward to receiving your revised manuscript.

Kind regards,

Zuowei Wang, Ph.D.

Guest Editor

PLOS ONE

Journal Requirements:

Additional Editor Comments:

I enjoyed reading this revised manuscript. I was originally “Reviewer 2”. In Feb 2024, the journal invited me to serve as the Guest Editor to handle this revised manuscript. Since then, I sent out the manuscript to multiple reviewers, and have recently received one review back. The review is provided under “Reviewer 3”.

The authors have provided the necessary information I asked in my previous review. I’d like the authors address the following issues before I can make a recommendation.

First, based on my own reading and Reviewer 3’s comments, it is critical that the authors explicitly acknowledge the potential impact of low reliability of WIAT-II (alpha = .62, barely acceptable) on conclusions. WIAT-II’s low reliability could well be the cause of 1) low correlation to NDRT, and 2) its failure to predict eye-tracking results. These alternative explanations need to be clearly discussed. The disattenuated correlation between WIAT-II and NDRT is actually not bad (.22/sqrt(.62*.75) = .54, which is much stronger, and closer to what one would expect to see as the strength of correlation between two reading comprehension tests. The argument of this paper -- reading comprehension tests can test different things – sets a much higher bar for what might be considered acceptable reliability of a test. Only acknowledging the relatively lower reliability of the tests in lines 673-676 is insufficient.

Second, please add correlation coefficients “overall accuracy on comprehension questions” (line 385) and both WIAT-II and NDRT scores. The former may be lower, due to WIAT-II’s low reliability.

Third, the introduction is nicely written. Adding subheadings between transition of different topics will be very helpful. Also related to subheading, line 384 “Data cleaning” should be moved to line 392, since the first paragraph in Results is not really data cleaning.

Fourth, lines 421-425, please add a citation and rationale for this approach. Intuitively, one would build the simplest model and add more effects to it until new effects are no longer significant.

Reviewers' comments:

Reviewer's Responses to Questions

**Comments to the Author**

1. If the authors have adequately addressed your comments raised in a previous round of review and you feel that this manuscript is now acceptable for publication, you may indicate that here to bypass the “Comments to the Author” section, enter your conflict of interest statement in the “Confidential to Editor” section, and submit your "Accept" recommendation.

Reviewer #3: All comments have been addressed

2. Is the manuscript technically sound, and do the data support the conclusions?

Reviewer #3: Yes

3. Has the statistical analysis been performed appropriately and rigorously? 

Reviewer #3: Yes

4. Have the authors made all data underlying the findings in their manuscript fully available?

Reviewer #3: Yes

5. Is the manuscript presented in an intelligible fashion and written in standard English?

Reviewer #3: Yes

6. Review Comments to the Author

Reviewer #3: This manuscript discusses how different comprehension measures may tap into different underlying aspects of comprehension despite sharing the same name. Researchers compare the extent to which the two comprehension measures are predictive of eye tracking data. Both qualitative and quantitative differences between the tests are discussed. The authors do an excellent job of reviewing the literature and framing the argument.

Overall, this study offers a unique contribution to the literature, especially given its use in an adult population of readers. While the Cronbach’s alpha levels were lower than desired, the others acknowledge this and offer a reasonable explanation (lines 378-382; though it would have been nice to have expanded on this). Additionally, the methods and results section were, for the most part, clearly written (though it was unclear to me what a “full random structure” means in terms of multi-level modeling—random slopes? Or random intercepts? Both?). The discussion section was thorough and addressed some of limitations associated with the study.

I feel the previous reviewers have helped the authors address most major issues. I have no major concerns with the paper and recommend it for publication.

7. PLOS authors have the option to publish the peer review history of their article (what does this mean?). If published, this will include your full peer review and any attached files.

Reviewer #3: **Yes: **Daniel P. Feller

---

## [Author Response · Author response to Decision Letter 2]

6 Jun 2024

Dear Dr Wang,

We would like to thank you and the reviewers for taking the time to review this manuscript. My co-authors and I have now addressed the comments in the text below and changes are indicated in the manuscript.

Kind regards,

Dr Charlotte Lee

Journal Requirements:

Additional Editor Comments:

I enjoyed reading this revised manuscript. I was originally “Reviewer 2”. In Feb 2024, the journal invited me to serve as the Guest Editor to handle this revised manuscript. Since then, I sent out the manuscript to multiple reviewers, and have recently received one review back. The review is provided under “Reviewer 3”.

The authors have provided the necessary information I asked in my previous review. I’d like the authors address the following issues before I can make a recommendation.

First, based on my own reading and Reviewer 3’s comments, it is critical that the authors explicitly acknowledge the potential impact of low reliability of WIAT-II (alpha = .62, barely acceptable) on conclusions. WIAT-II’s low reliability could well be the cause of 1) low correlation to NDRT, and 2) its failure to predict eye-tracking results. These alternative explanations need to be clearly discussed. The disattenuated correlation between WIAT-II and NDRT is actually not bad (.22/sqrt(.62*.75) = .54, which is much stronger, and closer to what one would expect to see as the strength of correlation between two reading comprehension tests. The argument of this paper -- reading comprehension tests can test different things – sets a much higher bar for what might be considered acceptable reliability of a test. Only acknowledging the relatively lower reliability of the tests in lines 673-676 is insufficient.

Second, please add correlation coefficients “overall accuracy on comprehension questions” (line 385) and both WIAT-II and NDRT scores. The former may be lower, due to WIAT-II’s low reliability.

Third, the introduction is nicely written. Adding subheadings between transition of different topics will be very helpful. Also related to subheading, line 384 “Data cleaning” should be moved to line 392, since the first paragraph in Results is not really data cleaning.

Fourth, lines 421-425, please add a citation and rationale for this approach. Intuitively, one would build the simplest model and add more effects to it until new effects are no longer significant.

Author response:

1. We have now included further discussion of how low reliability may impact the interpretation of the results on lines 670-677. We also mention this in the abstract. 

2. Correlation coefficients for accuracy on the eye tracking task and WIAT-II and NDRT comprehension scores have now been included on lines 396 - 398.

3. We have now added subheadings in the discussion so that the different topics are clearly signposted. We have also moved the Data Cleaning subheading to line 399.

4. We have now moved the citation for the model trimming procedure to the end of the sentence on lines 425 - 433 so that it is more clearly signposted and we have now included some clearer explanation of the model building/trimming method used. 

Review Comments to the Author

Reviewer #3: This manuscript discusses how different comprehension measures may tap into different underlying aspects of comprehension despite sharing the same name. Researchers compare the extent to which the two comprehension measures are predictive of eye tracking data. Both qualitative and quantitative differences between the tests are discussed. The authors do an excellent job of reviewing the literature and framing the argument.

Overall, this study offers a unique contribution to the literature, especially given its use in an adult population of readers. While the Cronbach’s alpha levels were lower than desired, the others acknowledge this and offer a reasonable explanation (lines 378-382; though it would have been nice to have expanded on this). 

Additionally, the methods and results section were, for the most part, clearly written (though it was unclear to me what a “full random structure” means in terms of multi-level modeling—random slopes? Or random intercepts? Both?). The discussion section was thorough and addressed some of limitations associated with the study.

I feel the previous reviewers have helped the authors address most major issues. I have no major concerns with the paper and recommend it for publication.

Author response:

1. We thank you for this comment and have now added further discussion of the low alpha for the WIAT-II on lines 670-677.

2. We have also added further explanation for the structure and trimming procedure for the LMMs on lines 425-433.

---

## [Editor Report · Decision Letter 3]

19 Jun 2024

The jingle fallacy in comprehension tests for reading

PONE-D-23-23595R3

Dear Dr. Lee,

We’re pleased to inform you that your manuscript has been judged scientifically suitable for publication and will be formally accepted for publication once it meets all outstanding technical requirements.

Kind regards,

Zuowei Wang, Ph.D.

Guest Editor

PLOS ONE

Additional Editor Comments (optional):

I can now recommend publication of this manuscript.

This paper provides a comprehensive review of the existing literature on the differences of existing reading comprehension measures before it connects to the eye-tracking literature to set the stage for the study. The logic is clear and compelling. It used a convenience sample of college students. The sample size was not big, and the distribution of ability in this sample was not ideal due to a lack of students in the middle of the distribution. The authors had to adjust the scoring and testing method to reach an acceptable test reliability. The low reliability of reading comprehension measures decreases my confidence in the replicatability of this study. However, the eye-tracking results seem robust and consistent. I hope the study will encourage more future research on this matter. Given the inter-disciplinary nature of this journal, I believe this paper, once published, will be interesting to the journal's audience.

Table 11 has a typo (line 564). Table 10 is missing two lines. I leave formatting issues to the authors and the journal's production office.
---

## [Editor Report · Acceptance letter]

25 Jun 2024

PONE-D-23-23595R3 

PLOS ONE

Dear Dr. Lee, 

I'm pleased to inform you that your manuscript has been deemed suitable for publication in PLOS ONE. Congratulations! Your manuscript is now being handed over to our production team.

Kind regards, 

on behalf of

Dr. Zuowei Wang 

Guest Editor

PLOS ONE